# Ensuring sexual and reproductive healthcare services amidst a pandemic: Experiences from health workers in Lima, Peru

Camila Gianella[1]*, Leonardo Cortez[2], David Beran[3], Maria Amalia Pesantes[2]

**1** Department of Social Sciences, Pontificia Universidad Católica del Perú, Lima, Perú, **2** CRONICAS, Centre of Excellence in Chronic Conditions, Universidad Peruana Cayetano Heredia, Lima, Perú, **3** Division of Tropical and Humanitarian Medicine, University Hospitals of Geneva and University of Geneva, Geneva, Switzerland

* gianella.c@pucp.edu.pe

## Abstract

Nowadays there is an emerging interest on health system resilience capacity during emergencies as the one created by the COVID-19 Pandemic. This article contributes to this emerging field of studies by analysing the impact of the state´s policy responses COVID-19 (as lockdowns) on the Peruvian health system, specifically on the delivery of non-covid services, sexual and reproductive health services, and describe the strategies deployed by health workers to adapt to the COVID-19 crisis in Peru, a country that have been dramatically impacted by the pandemic. The article, based on the analysis of depth interviews with 11 health workers and one health supervisor working at sexual and reproductive health services at public health services Lima during 2020 and 2021, describe how pre-existing conditions of the health system (as poor infrastructure and deficit of human resources) magnified the negative effects of the measures taken to control de pandemic, undermining the "resilience" of the health system.

**Data Availability Statement:** The qualitative data collected in this research includes sensitive data that could facilitate the identification of the participants. Because of this and the commitment

## 1. Introduction

The aftermath of the COVID-19 pandemic has shown the importance of building resilient health systems that are able to cope with and respond to a crisis in parallel to guaranteeing the maintenance of their core functions [1]. Evidence shows that in most countries, public health measures to control the spread of COVID translated into closing health facilities, restricting access to patients due to lockdowns, and reducing non-COVID visits. In addition, there was the perception that hospitals and health centres were "COVID-19 infection-risk places" [2–4]. Less is known about the impact of these COVID-related factors on the health workforce and their activities in continuing to deliver primary health, even in countries severely impacted by COVID, such as Peru.

Some studies have shown that, in addition to public health measures to prevent the spread of COVID-19, health workers were reassigned to provide COVID-19 care, negatively impacting population health [4–7]. Redistribution of workload and responsibilities, coupled with the

made by the research team to guarantee confidentiality, expressed in the informed consent form, records and transcripts are not available to the public. In case researchers are interested in accessing the transcripts for research purposes, they could contact CRONICAS Center of Excellence in Chronic Diseases at ceec@oficinas-upch.pe. Because of the characteristics of the research, and to guarantee respect to the interviewees (i.e. avoid misinterpretations), it is expected that researchers asking to access the data speak Spanish and have knowledge of the Peruvian health system.

**Funding:** Project funded to MAP by the HRP Alliance, part of the UNDP-UNFPA-UNICEF-WHO-World Bank Special Program for Research, Development and Research in Human Reproduction (HRP), a co-sponsored program run by the World Health Organization (WHO). WHO). This work was also supported by the Alliance for Health Policy and Systems Research, Science Division of the World Health Organization, the Center for Perinatology, Women's Health and Reproduction (CLAP) of the Pan American Health Organization and the Center for Reproductive Health Research of Campinas (CEMICAMP), Brazil. These funds were received by Dr. Jaime Miranda and Dr. María Amalia Pesantes. The funders had no role in study design, data collection and analysis, decision to publish, or preparation of the manuscript.

**Competing interests:** The authors have declared that no competing interests exist.

challenges of delivering care under such critical conditions, affected the mental health of front-line health workers dealing with COVID-19 [8–13]. This situation was intensified by a poor re-structuring of the health system and health care supply, including the organisation and availability of non-COVID-19 services [3, 5, 14–18]. Despite the emotional toll and increased workload health workers experienced, they continued working. Our paper aims to contribute to this emerging field, by presenting health workers' accounts regarding the changes introduced in Sexual and Reproductive Health (SRH) and the impact of COVID-19 on their capacity to ensure continuity of care in primary health care facilities in Lima, Peru and discusses the importance of looking beyond the crisis to guarantee continuity of care for other essential health needs.

## 2. Background

### 2.1 COVID-19 in Peru

Peru was one of the first Latin American countries to adopt a mandatory quarantine at the national level to stop the spread of COVID-19 [19]. On March 15, 2020, with no reported deaths, the government issued Supreme Decree No. 044–2020,-PCM declaring a nationwide state of emergency. This involved closing borders, curfews, closing schools, universities, and generally restricting all non-essential activities or services, including non-emergency primary health care services. Some of these measures were maintained for more than 100 days.

Before the pandemic, the Peruvian health system faced a shortage of human resources and a precarious infrastructure, resulting in barriers to access to health services, including antenatal care. Before the COVID-19 pandemic, 91.2% of Peruvian women reported problems accessing healthcare services when needed [20]. Furthermore, geographic accessibility had already been identified as one of the main barriers to maternal care before the COVID-19 pandemic. In addition to limited access to care, before the COVID-19 pandemic, the limited number of health facilities was already a significant challenge in Peru.

By January 2020, according to the Ministry of Health, 77.8% of primary health care services needed more capacity, expressed in the precariousness of the infrastructure ad obsolete, inoperative or insufficient equipment [21]. The COVID-19 pandemic forced Peruvian health authorities to take measures that undermined an already weaken system. To protect the health workers, in March 2020, the Peruvian State ordered health workers with comorbidities to take leave or work remotely (DU 026–20 from March 2020) [22]. To address the human resources shortage and increasing demand for COVID-19 services, Peruvian health authorities reorganized and reassigned the available health workforce to cover COVID-19 services. New staff was hired to work exclusively on the COVID-19 services [23]. Additionally, primary health care facilities were closed, affecting the provision of sexual and reproductive health services, including services to regular antenatal control. No evident efforts were taken to guarantee the follow-up care of pregnant women. As a result, regulations addressing pregnant women's health during COVID-19 were very scarce. Declarative statements with regards to securing access to regular health services at the primary health care level, such as vaccination, anaemia prevention, and antenatal control, were made [24]. Yet, none of those statements translated into clear guidelines to reallocate health workers, respond to the shortage of health professionals, or strategies to adapt the services to the new emergency context. In April 2020, to protect health workers and guarantee access to health services, the government announced the possibility of telemedicine to ensure the continuity of care for non-COVID-19 conditions. However, the action did not consider the poor communication infrastructure of both health facilities and patients: eight out of ten primary care centres do not have Internet [25], only 84.4% of poor households have access to a cell phone, and only 7.1% have access to the Internet

[26]. Thus, this measure did not help cover the demand for health services at the primary health care level.

According to the "*Colegio de Obstetras del Peru*" (Peruvian Professional Association of Obstetricians), the number of midwives working in public health services before the pandemic (around 17,000) was already insufficient. During the pandemic this worsened given that about 40% of midwives stopped providing in-person health services as many had a comorbidity that put them at risk and thus prevented them from working at a health facility or because they got infected with or died from COVID-19 [27]. A 2021 press release by the "*Colegio de Obstetras del Peru*" [28] stated that "As of February 25 of this year, 3123 obstetricians have been infected with COVID-19, as a result of their work, of which 41 have died, 15 are hospitalised, and six are in ICU fighting for their lives." The impact of this on access to antenatal care remains unknown. Still, in a country where 83.8% of women report antenatal care with an obstetrician [29], there are reasons to believe it has been a critical factor in the increased number of maternal deaths in 2020 and 2021 [30]. A recent report estimates that the number of appointments in public sexual and reproductive health services in Peru during the pandemic has decreased by 14%. [27, 30], but little is known about the quality of the services that were offered.

## 2.2. Availability of sexual and reproductive health services in Peru

Since the mid-1990s in Peru, significant efforts to increase the coverage of modern family planning methods, reduce fertility rates among Peruvian women, and prevent maternal mortality have been implemented. Although these efforts have enabled the reduction of maternal mortality (from 185 per 100,000 live births in 1994 to 93 in 2015) [31] and increased the coverage of the use of modern contraception methods (from 50.4% of women between 15 and 49 years old in 2000 to 55% in 2020) [32, 33], they were not linked to a comprehensive universal health care policy including access to critical care [21].

## 2.3 Methodology

This qualitative study is based on in-depth interviews with health workers at sexual and reproductive health units in public health facilities in Lima, Peru. The interviews were conducted during April and August 2021 with 11 health workers and one health supervisor to explore their perceptions and experiences with sexual and reproductive services during the pandemic (n = 12). We asked about the impact of COVID-19 on sexual and reproductive health services and the adaptations they implemented. This question elicited detailed descriptions of the various actions they had put in place during the pandemic, as well as their feelings and assessment of the responses from health authorities. Conversations allowed us to explore the adaptations health workers implemented to continue providing SRH services during the lockdown and amidst the governmental regulations regarding COVID-19 prevention in a city greatly affected by the pandemic.

We conducted the interviews through videoconference calls. Interviewees worked in public hospitals (3), a maternal health centre (2), primary health care facilities (3) and health centres (3). Health professionals were either primary care physicians (two) or midwives (nine), and the time working as a healthcare worker in a public health facility ranged from 5 to 20 years. Interviews lasted an average of 42 minutes. Only one included interviewing two health workers simultaneously. All interviews were conducted by two authors of this paper, both trained in anthropology and with extensive experience in qualitative research around health issues. Connectivity was a problem with some participants, but overall, the interviews were conducted without significant interruptions. Most health workers were at the health facility at the time of the interview, while others were at home. In one case, the last question was answered via

WhatsApp through a recorded message because we lost connectivity, and the participant wanted to share her views on the topic we were discussing. This study received ethical approval (IRB approval number 200299 from XXX ethical review board).

Recorded audio data from individual in-depth interviews were subjected to careful transcription. Transcripts were read and re-read by three authors who held three meetings to discuss the key themes that were emerging from the data and that showed the impact of the pandemic in the services they were providing, the strategies they were developing to face such changes and the emotional impact of the pandemic. Two of the co-authors coded the data using Atlas-ti 8.0 along those three themes. The quotes from those themes were shared with the other co-authors and were then discussed on one meeting to find the interconnection between the themes as well as the similarities and differences between the health workers' experiences and possible explanations for such differences. Although we did not use an specific framework of analysis, our analysis and interpretation was informed by the debates about the role of human resources in ensuring strong and resilient health systems, and the importance of documenting changes at the micro-level to understand macro-level processes. This strategy was helpful as it enabled us to encounter unexpected insights [34] and identify the most relevant examples.

## 3. Results

This section presents the results from the interviews in the following sections: the health system's pre-existing weaknesses, the impact of the pandemic on the health system and health workers, and finally, the strategies implemented by health workers to continue providing sexual and reproductive health care and the capacity of the health system to adapt procedures to the emergency setting.

### 3.1 Existing weaknesses in the health system

Peru's health system´ was precarious even before the pandemic: limited number of physicians and nurses, 133 beds in intensive care units for a population of over 25 million people, poor implementation of universal health coverage, and a fragmented structure that makes it nearly impossible to refer patients between public health facilities, private health facilities and facilities managed by the social security system [35]. This precarity limited its capacity to cope with the pandemic and to ensure health workers felt safe providing pre-natal care that requires proximity with the pregnant woman. Providing antenatal care was particularly problematic in the early months of the pandemic, when there was a poor understanding of the various transmission routes, and vaccination was unavailable. In addition, one major challenge faced by health workers in non-COVID-19 services, such as SRH services, was the infrastructure limitations: namely, the lack of space. One participant explained:

"It has been a challenge to adapt healthcare provision in the context of the pandemic, especially if you consider the infrastructure of our health facility. I believe that in 2011 this health facility was built as a small two-floor building, with minimal space, around 20x20 square meters, and the architectural design was [not appropriate for a health facility]. We have slowly introduced changes in the building to improve our services, but to be honest, our facility does not have a good design, even worse now that we need to provide care for COVID patients." (Interview 2)

Other interviewees also mentioned that the allocation of certain rooms or areas (in the case of Hospitals) exclusively for COVID-19 patients meant that maternal care services had to be

provided in inadequate spaces. In some facilities, the physical space of outpatient services like SRH was allocated to COVID-19 patients, resulting in even less space for non-COVID-19 services. This was inadequate for treating COVID-19 patients, but also perhaps not safe to provide care for non-COVID-19 patients:

"... they took away a lot of [our] beds, we have two rooms, one on the right and the other on the left. The entire third floor is the gynaecological area in the hospital. [During the pandemic] beds were removed, a whole ward was removed, the bed area was removed, and sent to the COVID area. They changed the emergency area. We used to provide care in the emergency area, but they transferred us to the trauma area, where the doctor's offices are. ... so we had to see patients in the offices, and it was weird because we saw patients inside offices with small doors, sometimes we saw patients just like that in the hospital corridor." (Interview 9)

### 3.2 The direct impact of COVID-19 on the supply of non-COVID-19 services

One feature of the COVID-19 response described in the testimonies was the lack of planning. Health workers received a general order to close down some services, reorganize the referral system at the PHC level, and reallocate health workers to support COVID-19 efforts. There were no clear indications on how, under these new circumstances, non-COVID health services should be organized, ensure patient follow-up, or guarantee regular access to drugs or supplies (like contraception).

"At the beginning [of the pandemic], we only received instructions from the Ministry of Health regarding general medical services, and only for some things because even regular immunisations for children were suspended. Until then, obstetric services were also on hold, but the staff decided not to interrupt those services. I think it was only during the second half of March [2020] that services were not available. Later, services resumed, and the strategy the obstetricians developed was [given that the health facility only has two obstetricians]... "well, let's do three 12-hour shifts a week" to cover at least the shifts in the morning and afternoon" (Interview 1)

Sick leave and reallocation of health workers also implied that some services were left without the more experienced physicians who were "sent home" because their age put them at risk, but also without health care technicians. This forced health workers to assume new responsibilities and duties, affecting their capacity to provide all the services they used to provide. For example, one OBGYN explained how not having more experienced doctors around affected the services they offered in the birthing area of the hospital:

"[T]he older doctors had to leave, so we were in charge, with little or not as much experience. So obviously, there are people with many more years of experience who were also with us. Still, the people with more experience had to stop coming to the hospital [to protect them from getting infected]. So, we were learning what to do in "real time" as the patients came, and we evaluated them...." (Interview 9)

Another nurse explained:

"Another problem has been [the government´s mandate of] sending the workers over 65 to their home. The average age of our workforce is high, especially the technical staff, and they

had to go home. We were left without technical staff. The few remaining were the youngest and were sent to work on COVID. Now the offices do not have technical staff. Not having all these support staff meant we had to do everything, including triage, and we couldn't do everything. For example, when the pandemic started, when I placed an IUD, I had nobody to wash the instruments because the technical staff had been assigned to the birthing area and did not want to help. This is a problem that we have until now. Sometimes to introduce an IUD, I have to search for the nursing staff to take care of all that because we no longer have the technical staff to support us. If I am going to introduce an implant or I am going to remove an implant, I can't find a nurse technician to wash the materials" (Interview 6)

In the case of sexual and reproductive services, the closing of small health posts and health centres created an increase in the catchment area of non-COVID health facilities in addition to distance-related barriers for users. Non-COVID health facilities had to provide care for users of other facilities that were either closed or had become facilities dedicated to COVID-19 services: "Usually, this hospital gets referrals from 44 health facilities, but when the pandemic started, we had to accept patients from other areas [of the city], and now we receive patients from around 66 health facilities." (Interview 8)

The health services that continued to provide care (such as maternal health services) needed the capacity to guarantee adequate coverage. In addition, the less available health workforce, combined with additional biosecurity control measures and restrictions regarding the number of people at the health centre at the time, made it impossible to keep providing the same level of care.

"So the volume of patients we can see has been restricted; we have two obstetricians, four of us are doctors, but only three of us are working because one has not been able to return since March of last year because he has a comorbidity and because he is old. . . The same happened with the nursing staff, with technical staff [the number of available nurses and nurse technicians has been reduced], there are still seven or eight workers doing telework, which also reduces health workers' supply a bit. . .." (Interview 1)

### 3.3 Fear among health workers

COVID-19 also generated fear among health workers. They were concerned about getting infected. This affected the quality of healthcare provision. The supervisor of sexual and reproductive health services (whose responsibility is to visit health facilities to ensure quality standards are being followed) explained that many of the midwives were concerned about getting infected and stopped offering contraceptives that required close contact with patients, such as implants, injections or IUDs:

". . .the midwives were afraid of catching it [COVID-19], as simple as that. And until now, many refuse to put an implant. . ., to do a Pap smear. . .. Then [2020] there was fear and there is still that fear." (Interview 12)

In addition, health workers expressed that they felt that their concerns and in general their mental health was neglected by health authorities. To some extent, the system has ignored that health workers have been also affected by the pandemic, that they also are afraid and have experienced loss. The focus has been on productivity, to continue reaching the quantitative goals stated for the health facility sexual and reproductive health services by health authorities, and this has an impact in the health workers' morale:

"It seems to me that the Regional Health Office is not very interested in its workers, the only thing it is interested in is that the worker treats patients and meets the goals that they must meet, and it does not see the emotional side of the workers. If the worker suddenly lost two or three relatives and is depressed or is afraid of being infected. . . it is an everyday dilemma. Nobody has told us: "they need psychological therapy", the emotional side [of the work] is not being taken into account (. . ..) The goals continue, they are the same targets as before, they have not reduced the targets since there is no attention. Health authorities are always trying to demand us to comply and fulfil the targets and well, right now people are a little reluctant and they no longer respond to those demands, and they are not concerned by the threats anymore. . ." (Interview 4)

The importance of reaching quantifiable targets, a characteristic of the Peruvian health system, continued despite the complex environment in which they were working.

### 3.4 Responses from the heath workers: Adapting to the new reality of COVID-19

Despite all the challenges described in the previous sections, another common theme in health workers' narratives was their concern to ensure the continuity of sexual and reproductive health services. Ensuring continuity of care required innovation but oftentimes also more work or even money that had to come out of their own pockets. Initially, sexual and reproductive health care providers independently adapted to meet with those patients who just needed access to contraceptives, via telephone or video call. However, video calls were only sufficient for one aspect of the chain of health services required: consultation, counselling and prescription of the most appropriate family planning method. The challenge was to ensure the patient actually received the contraceptive, and this required close coordination with the person in charge of supplies at the pharmacy located at the health facility. This, as the following quote shows, depended on the capacity of the midwife to coordinate with the pharmacy at the facility and their willingness to invest their own money to ensure the prescription reached the pharmacy:

"[After the online consultation] I would write the prescription at home, I would send the photo to the pharmacy lady, the pharmacy lady knew that the patient was coming to pick up the medication. She also sent her the photo and the supplies reached her through the gate. It was a way of not leaving them unprotected without a [family planning] method, but even for that there were drawbacks because at some point, the lady from the pharmacy told me:

- When are you going to bring me the prescriptions? I need them physically.

- I will send them over the weekend.

- No, you have to send them on a daily basis.

- I can't send you the prescription every day.

Once a week I could send her the paper prescriptions with a motorcycle, a delivery, something like that. But then she told me she couldn't work like that either. And since I was working from home, they didn't pay me more than the basics, they didn't pay me night shifts, they didn't pay me other benefits that we used to have before the pandemic. . . [covering the prescription delivery costs] didn't suit me either. And so we are now cutting that system also. (Interview 12)

This example shows that in the face of inventiveness, health workers have also faced bureaucratic barriers to implement strategies that allow women to get required medicines or supplies on time such as requiring paper prescriptions issued in the context of an online consultation. This shows the rigidity of the system in adapting to an emergency where online consultations are enabled but online prescription not.

Across the testimonies, we found that health workers tried to adapt to the new circumstances. One main concern was to organize the service to secure the patients' follow-up. One strategy that emerged was the use of mobile phones to send patients reminders of their appointments or information regarding where to go for the delivery. This task has been carried by the obstetricians on duty, and/or the staff working remotely (in sick leave).

> "So what we did as a hospital, with our colleagues and the health personnel in general who switched to remote care, is that . . . they were the ones responsible of speaking with our pregnant users, they called all the pregnant women who were scheduled for outpatient consultation, monitored them, asked them how their health was.(. . .) In case they had Covid or even if they did not, we told them to come through the hospital's emergency service, or we told them to go to another hospital and do the prenatal monitoring up to the moment of delivery. . ." (Interview 8).

> "When a pregnant woman comes, I always ask her for her phone number. Actually I ask for it from everybody: pregnant women, users of family planning services and also the ones who come for a pap smear, so when her appointment is scheduled at least for those who have an appointment the following day, I'm writing to them the previous day: 'look Maria, tomorrow is your prenatal check-up, I'll wait for you at 7 in the morning or I'll wait for you at 2 in the afternoon', and they will answer me, they tell me, 'Miss, I can't in the morning, I can in the afternoon', 'Ok, come in the afternoon'. And then I write down more or less how many women are going to come in the morning and in the afternoon. . ." (Interview 2)

Using mobile phones to follow up pregnant women is not a new tool. Some of the interviewees said that, before the pandemic, they already used text messages when one of the pregnant women they were seeing stopped coming to their appointments. However, during the COVID-19 pandemic they adapted the tool to follow-up on all pregnant women (in general), and in some cases also to follow up the users of modern contraception, to send them reminders. The flipside of this strategy, is that health workers' cell phone numbers became *vox populi* and they were on call all the time, to make appointments, to ask about when they could go to get a new contraceptive, etc.

Cell phones and video cameras also became a tool to communicate with relatives during the delivery. Due to biosecurity measures, relatives, as partners, are not allowed to be in the ante natal control, or the delivery room as before the pandemic. To create an emotionally safe and a friendlier environment for everybody, health workers have been using their phones to call the relatives, and to send photos of the new-borns.

> "One thing that we are testing is "the delivery photo". What we are doing is, when one of the midwives is in the delivery room, the one in charge of the delivery calls the other two midwives to take a picture. For example, imagine it is my turn in the hospital this month so my colleague calls: "Carola!" and I run with my phone and I take the photo of the moment when they put the baby in the mothers' chest after the delivery. The baby is still wet and the mom is still sweating and "Pam!" I take a photo of her, and I send the picture to the physician or midwife who has seen her throughout her prenatal care check-ups, and I also send

the picture to a family member she chooses. At that moment we ask her "Mrs, what is the fathers' number!" And send the picture to him. So, he is not present, he is not inside in the delivery room, but he is living that moment. They afterwards, make that photo viral, they send it to her entire family, to all of her networks, I don't know. I am sending the picture with the consent of the paediatrician and the mother. "Ma'am, do you want me to take your picture?" "Yes" "Ok". We wanted to also print the picture, but for logistical reasons we have not been able to do it, but at least with the virtual picture we feel that we are giving the patient some confidence: "push, push, I'm going to take your picture." We are testing [this strategy] to see how it works." (Interview 2)

These efforts to adapt the system were implemented despite the fear health workers had about getting infected. According to one interviewee, such fear not only decreased the number of implants and family planning shots, but even increased the number of C-sections.

"[I remember that early on in the pandemic] a study had come out that one could be infected through faeces. And it turns out that when one gives vaginal birth, some women during labour, defecate and one is in contact with the faeces. Then, as some midwives related vaginal birth with an increase chance of infection, they decided to switch to caesarean section. . . . to be honest, everything was new, we were seeing how we could deliver babies [without getting infected]. Even the COVID management scheme was not very well defined at the beginning. The WHO said one thing, but in practice another treatment worked. Everything went little by little." (Interview 11)

This last quote shows the limited information health workers in charge of sexual and reproductive health services had to continue providing care, and points to some of the impacts it might have had on delivery experiences and family planning usage during the pandemic.

## 3.5 An ossified health system

The pandemic has shown that the Peruvian health system, has no resilience capacity to adapt and respond to crisis and new demands. Each health facility was solving their problems differently, depending on the staff's capacity to innovate and collaborate. Health authorities continued demanding health workers to reach pre-pandemic target numbers (prenatal care visits, contraceptive distribution, counselling sessions, among others) while overlooking the deep emotional needs of healthcare providers who felt abandoned. In several interviews, healthcare workers shared stories showing lack of support from health authorities even when they were facing the loss of their colleagues. One of the health workers shared her feelings when the head of the health facility, a dear colleague and friend, died in April 2020 from COVID-19.

"[Despite being working from home because of my age and my blood pressure] I personally came to the meeting organized by the health workers [in his honor]. They organized a funeral mass in the health facility on his behalf. [A person from the regional health directorate came] and I told him that he was not welcomed because they never bothered to say: "there is no doctor there in that health facility, their doctor died." They never came, they never asked anything. So the mass was only with his family and us, his colleagues with his ashes, celebrating a mass." (Interview 10)

On the other hand, we learned that the strategies developed by health workers, were therefore financed by health workers. To provide follow-up care, health workers used their private phones, and covered with their own money, the cost of the calls, video calls and WhatsaApp or

SMS messages. In April 2020 the Ministry of Health announced the use of telemedicine as a tool to improve access to health care during the pandemic, however, funds have not been allocated, health workers have not received the required equipment (as cell phones with a camera and an internet plan), and others only got a phone for a short period of time as this midwife describes:

"But it turns out that they gave us that cell phone at the end of September, [we kept it] October, November. In December they asked us to return the cell phone because the program that had donated had ended and the telephone lines had no more funds in them. There was no longer a telephone line. And then we had to return it. And I don't know what happened to those cell phones. But in general I have worked all the time with my cell phone. Well, the patients already memorize my number, now everyone calls me." (Interview 10)

As this quote shows, the health system has relied to some extent on health workers' personal equipment, and internet plan despite the fact many of health workers experienced wage cuts due to the fact that they could not do additional shifts (and earn extra income) when they were working remotely. Furthermore, they have been organizing online meetings and trainings without an analysis of the conditions under which health workers worked from home. As schools were closed, and education was remote (schools in Peru closed from March 2020 until March 2022), many health workers shared their cell phones, laptops or tablets and internet connection with their children. This situation reduced health workers' capacity to participate on the trainings, or meetings as this midwife describes:

"I think it would be fair to say that we are not as trained or maybe that we are not being given as many trainings as before. I see that all the time there are a lot of zooms! Zoom of this, zoom of the other, but to be honest I don't have the time to join them because you have to see our reality. Before we use to say: "you have your issues at home and they stay there, but you come to work and you have to work, you have to fulfil your professional responsibilities. But now, we are living in another reality. I sent my three daughters to school and I could go to the trainings, I could go to a course, I could go to work, do overtime, do "x" different things. I cannot do that right now. Not now. Now I don't even watch television in the morning so as not to slow down the internet connection because my daughters are at home. Before, I would send them their lunch box and that's it. Now, I'm in the kitchen all day! So, the reality of each person has changed. At least I can't be logging in all day, every day at the zoom, at the zoom. No way!" (Interview 5).

The impact of health workers with school-age children and small children at home is yet an unexplored issue but as this quote shows it certainly had an impact in the opportunities to attend online trainings aimed at professional development.

## 4. Discussion

The Peruvian government was praised for its rapid response to COVID-19, and for the implementation of a general lockdown. Almost no observations were raised at the time regarding the potential negative impacts of closing down health services such as sexual and reproductive health services, nor the effects of demanding health workers to accomplish pre-determined unrealistic goals. Undoubtedly, emergencies such as COVID-19 have a negative impact on the capacity of health systems to continue providing regular care as they create unexpected pressing needs and pressure on health systems. However, across the globe health systems have shown different levels of resilience, understood as the capacity of health systems to prepare for

and effectively respond to crises, maintain core functions when a crisis hits, reorganise if the conditions require it and, more importantly, absorb and recover from shocks and stress informed by lessons learned during the crisis [36].

There is an emerging corpus of literature showing the adaptability and capacity of health systems to reorganize [37]. In many cases, such adaptability was the result of health workers' actions within "non-COVID-19" services to ensure the availability of regular health services. These studies, albeit scarce, suggest that even in an emergency situation with major pressures and constraints, health workers developed innovative strategies to adapt their services during pandemic times [38] to continue caring for their patients. Such strategies contain elements that are key when speaking about health systems' resilience which, implies not only the capacity to adapt to the new critical situation but also to absorb and recover from shocks and stress, without major negative consequences. Such studies argue that adaptability without robustness is not resilience.

This article presents health workers' accounts regarding the changes introduced in SRH and the impact of COVID-19 in their capacity to provide SRH services during the first years of the pandemic. Our analysis recognizes that while, health workers are at the core of resilience of health systems, health systems are much more than their health workforce. The number of health workers, as well as their preparedness, motivation, and engagement, are essential for the functioning of health systems and health outcomes in regular and emergency settings [1, 39]. In the case of SRH services as the ones included in this study, pre-existing conditions of the health system (poor infrastructure and deficit of human resources) magnified the negative effects of the measures taken to control de pandemic, undermining the resilience capacity of SRH services. These pre-existing conditions (that explain COVID-19 infections among health workers), coupled with the increasing number of new infections among the general population, created fear among OBGYN, which in turn decreased the number of medical appointments that were made available for users, in addition to a reduction and pause of specific services that complement prenatal care in Peru such as laboratory exams, nutritional screening and treatment [40]. The Peruvian health system has shown a lack of capacity to react to the needs of non-COVID-19 health care users, which have amplified the negative impact of the pandemic on population health and is probably behind the spike of 12% in maternal deaths in comparison with 2019 [30].

Our analyses also show the difference between the resilience of the health system and the resilience of the health personnel. Despite the adverse context, the health system rigidity, the lack of support for their mental health needs, and the fear of a disease that was killing their colleagues, health workers have shown their commitment right from the early stages of the pandemic. Furthermore, in the middle of a chaotic situation when there were no protocols on how to follow-up their users, they independently developed strategies to follow-up their patients, adapting their work to the new context (as forced health leaves, or restrictions in the number of appointments). However, our analysis also shows that the rigidity of the health system, paired with an excessive focus on numerical targets, jeopardizes health workers' capacity to adapt to new and challenging contexts as emergencies. Health workers have been feeling mistreated and overburdened. They were under the pressure of fulfilling targets designed stablished in pre pandemic times. In the context of a long-term crisis, as the one created by the COVID-19 in Peru, this is having major negative impacts on the health workforce.

## 5. Conclusions

The COVID-19 pandemic has left important lessons that must be considered when designing policies to strengthen health systems and prepare them for future crises. The performance of

the Peruvian health system during the pandemic shows the weakness of having a SRH services working under "acceptable" conditions in terms of a deficit health workforce and poor infrastructure.

The pandemic has unveiled a rigid health system and the perverse consequences of target-oriented SRH services. Health systems need to be assessed for their capacity to adapt to challenging circumstances and for their capacity to innovate, and not mainly by their capacity to deliver indicators and control processes. This requires a different way of thinking and engaging the workforce, recognizing their resilience and their commitment to patients' well-being.

## Author Contributions

**Conceptualization:** Camila Gianella, Maria Amalia Pesantes.

**Formal analysis:** Camila Gianella, Leonardo Cortez, Maria Amalia Pesantes.

**Funding acquisition:** Maria Amalia Pesantes.

**Investigation:** Leonardo Cortez, Maria Amalia Pesantes.

**Methodology:** Camila Gianella.

**Writing – original draft:** Camila Gianella, Maria Amalia Pesantes.

**Writing – review & editing:** Camila Gianella, Leonardo Cortez, David Beran, Maria Amalia Pesantes.

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
