## [Decision Letter · Decision Letter 0]

6 Sep 2023

PGPH-D-23-01179

ENSURING SEXUAL AND REPRODUCTIVE HEALTHCARE SERVICES AMIDST A PANDEMIC: EXPERIENCES FROM HEALTH WORKERS IN LIMA, PERU

Dear Dr. Gianella Malca,

Thank you for submitting your manuscript to PLOS Global Public Health. After careful consideration, we feel that it has merit but does not fully meet PLOS Global Public Health’s publication criteria as it currently stands. Therefore, we invite you to submit a revised version of the manuscript that addresses the points raised during the review process.

Please note that we have only been able to secure a single reviewer to assess your manuscript. We are issuing a decision on your manuscript at this point to prevent further delays in the evaluation of your manuscript. Please be aware that the editor who handles your revised manuscript might find it necessary to invite additional reviewers to assess this work once the revised manuscript is submitted. However, we will aim to proceed on the basis of this single review if possible. 

We look forward to receiving your revised manuscript.

Kind regards,

Miquel Vall-llosera Camps

Staff Editor

Journal Requirements:

1. Please provide additional details regarding participant consent. In the ethics statement in the Methods and online submission information, please ensure that you have specified (1) whether consent was informed and (2) what type you obtained (for instance, written or verbal, and if verbal, how it was documented and witnessed). If your study included minors, state whether you obtained consent from parents or guardians. If the need for consent was waived by the ethics committee, please include this information.

Reviewers' comments:

Reviewer's Responses to Questions

**Comments to the Author**

1. Does this manuscript meet PLOS Global Public Health’s publication criteria? Is the manuscript technically sound, and do the data support the conclusions? The manuscript must describe methodologically and ethically rigorous research with conclusions that are appropriately drawn based on the data presented.

Reviewer #1: Yes

2. Has the statistical analysis been performed appropriately and rigorously?

Reviewer #1: N/A

3. Have the authors made all data underlying the findings in their manuscript fully available (please refer to the Data Availability Statement at the start of the manuscript PDF file)?

Reviewer #1: No

4. Is the manuscript presented in an intelligible fashion and written in standard English?

Reviewer #1: Yes

5. Review Comments to the Author

Reviewer #1: Overall: The paper discusses how COVID-19 affected health worker capacity to ensure continuity of sexual and reproductive health care in Lima, Peru, given the changes introduced by the government to deal with the pandemic. The main argument of the paper appears to be that these impacts showed how the Peruvian health care system is ossified and that future efforts to strengthen it must go beyond mere ‘survival mode’ to ensure a robust system.

The paper does contain several grammatical errors which could be addressed by having a native speaker do a final edit to tighten the language and check for any English stylistic departures or grammar mistakes. The paper could also benefit from consistency in placing citations either inside or outside various punctuation marks, as well as consistent use of commas and semicolons. The paper is still quite understandable despite this.

Intro & Background: Authors clearly lay out the purpose of the paper. They provide evidence that the number of active midwives, already insufficient, declined during the pandemic, potentially becoming a factor contributing to increased maternal mortality during the pandemic. I have no changes to suggest.

Methods: Data analysis wasn’t entirely clear. It seems that a thematic analysis approach was used, given the citation and the statement that emergent themes were identified via group conversations. I recognize that the sample set is small enough that each IDI could be reviewed in its entirety, but I’d like more description of the process. How were the themes tracked? How were these themes compared across interviews? Did the authors use any grounded theory or framework analysis approaches in addition to the thematic analysis approach?

Also, the number of interviewers by site (10) does not match the numbers by profession (11), nor would this include the health supervisor (which brings total interviewed to 12). Please clarify.

Results: The first three paragraphs of the subsection on pre-existing weaknesses in the health system appear to fit better in the background section than here in the results section. I would suggest beginning this section instead with the sentence: “One major challenged faced by health workers … were the infrastructure limitations: the need for more space.”

Discussion: The discussion section suddenly introduces women’s rights, when this was not mentioned elsewhere in the paper as a central topic or motivation for writing the paper. Further, the results don’t focus on women’s accounts, nor how the pandemic measures affected women – the focus has been on the healthcare system and healthcare workers' experience within it. I suggest either mentioning women’s rights in the introduction as a rationale for the paper or eliminating the clause from the discussion. Mentioning women’s rights in the introduction will shape your argument differently, however: ‘In order to ensure women’s rights to healthcare, the government (or ministry of health) must attend to how COVID-19 measures affected on SRH services. This paper looks at those impacts…”

The discussion also reiterates the suggestion from the introduction about the cause of the maternal mortality spike during the pandemic. While suggestive, the interviewees do not mention this connection, so I would simply leave the suggestion in the introduction and not bring it back up in the discussion. If you want to discuss it in this section, then draw attention to how a quote from a specific interviewee you cited in the results section supports that speculation.

I particularly like the distinction between the resilience of the system as separate from the resilience of the personnel of that system. However, I would like to have seen some concrete recommendations flow from this distinction. For example, in the results section, the authors mention the flexibility of the personnel and their innovative creativity; the authors also mentioned that each facility was on its own and without communication from colleagues at other facilities and how they were adapting. A recommendation which follows from both these observations might be that the health system should implement a way for colleagues to share information on what they do at a regional level even during a pandemic lockdown situation; or that the health system survey how individual facilities adapted, be receptive to the changes which really demonstrated their utility, and then share those practices with others to encourage their uptake. In either case, I’d also draw out lessons which go beyond Peru and can be applied in other areas with functioning, but not robust, healthcare systems.

Conclusion: No comments beyond grammatical changes.

Grammar / typos:

Abstract

- “Peru, a country that has been dramatically impacted” (subject-verb agreement)

Intro

- “systems able to… guarantee the governance of their core functions” (subject-pronoun agreement)

- “dealing with COVID-19, [8-13] a situation that was…” (add indefinite article)

- “the Peruvian state ordered health workers with comorbidities to take health leave and work remotely” (“leave” is singular)

- “no clear efforts were taken to guarantee the follow-up with pregnant women.” (evidente = clear, obvious; eliminate article, change preposition)

- “declarative statements with regards to securing access” (use gerund, rather than infinitive)

- “Although however, these efforts have enabled” (although is sufficient)

Methods

- “grounded in in-depth interviews” (grounded in is the phrasal verb, even if it sounds awkward before “in-depth”; perhaps “based on in-depth interviews” is preferable to the authors?)

Results

- “challenge faced by health workers at non-COVID-19 services…” – do you mean at non-COVID-19 service sites? Or faced by health workers in non-COVID-19 services? (at a site vs in a service sector)

- “ but to be honest, our facility does not have a good design” (with comma, not semicolon)

- “In addition, we learned that in some… non-COVID-19 patients” (while not a run-on sentence, the sentence can be revised for clarity: “In some facilities, the physical space of outpatient services like SRH was allocated to COVID-19 patients, resulting in even less space for non-COVID-19 services. [etc]”

- “non-COVID health services should organize themselves, ensure patient follow-up, or guarantee access” (rare reflexive verb in English. Also, ‘patient follow-up’ is smoother than ‘patients’ follow-up’ – a rare case of a singular noun being used with a plural meaning in English, similar to la gente.)

- “drugs or supplies (such as contraception)” (I know this looks like 'tal como', rather than simply 'como'... otherwise use ‘like contraception’ here, but not ‘as’ without the ‘such’ in front of it.)

- “Sick leave and the reallocation” (again, use singular, as it has a plural meaning; insert definite article)

- Fear among health workers “stopped offering contraceptive methods that required” (not contraceptives methods; alternately, say: “stopped offering contraceptives that required…)

- Regional health office quote: “the only thing they are interested in is that the worker treats… and meets… and [they] do not see the emotional side of the workers.” (subject-verb agreement with they (the folks from the Regional Health Office), rather than the worker.) Later in the same quote: “They are always trying to demand that we comply and fulfil the targets”

- Responses from health workers: “the rigidity of the system in adapting” is better than “the stiffness of the system to adapt”

- “A rigid health system” or “ossified health system” (‘Stiffened’ in English can have the connotation of bracing oneself for a challenge or becoming resilient in the face of an anticipated challenge, which is not the connotation the subsequent paragraphs support. I suggest you carry this change through the discussion and conclusion sections, too.)

- “the health authorities continued to demand health workers to fulfil their ‘targets’ while…” (use a subjunctive verb, not an infinitive)

- Quote: “[Despite being working from home because]… They organized a [funeral/memorial] mass in the health facility…” (Specify that this was a funeral or memorial mass for clarity, and include the indefinite article before ‘mass’ to indicate the religious service.)

- “To follow up with their patients, they used…” (the phrasal verb needs ‘with’ for clarity)

- “despite the fact that many of health workers have experienced wage cuts…” (verb tense)

- “have been organized without an analysis of health workers’ home office conditions” or: “they have been organizing online meetings and trainings without an analysis…” (verb tense, plural possessive needs ‘ after the s, and home office is an adjective here rather than a noun, so remains singular – conditions is the noun.)

Discussion

- “In many cases… health workers’ actions within ‘non-COVID-19’ services to ensure…” (change preposition)

- Several areas need commas and semicolons either eliminated or moved. For example, I’d revise one sentence to look like this: “Despite the adverse context, the health system rigidity, the lack of support for their mental health needs, and the fear of a disease that was killing their colleagues, health workers have shown their commitment right from the early stages of the pandemic.”

Conclusion

- “must be seriously taken into account when…” (phrasal

---

## [Decision Letter · Decision Letter 1]

11 Jan 2024

PGPH-D-23-01179R1

ENSURING SEXUAL AND REPRODUCTIVE HEALTHCARE SERVICES AMIDST A PANDEMIC: EXPERIENCES FROM HEALTH WORKERS IN LIMA, PERU

Dear Dr. Gianella Malca,

Thank you for submitting your manuscript to PLOS Global Public Health. After careful consideration, we feel that it has merit but does not fully meet PLOS Global Public Health’s publication criteria as it currently stands. Therefore, we invite you to submit a revised version of the manuscript that addresses the points raised during the review process.

We appreciate your incorporation of previously suggested edits and suggest that you work with a copy editor to finalize this manuscript before resubmitting.

We look forward to receiving your revised manuscript.

Kind regards,

Laura Miniea Hoemeke, DrPH

Academic Editor

Journal Requirements:

Additional Editor Comments (if provided):

Thank you for addressing most of the comments and suggestions provided on the first draft of this manuscript. Please note that additional copyediting is needed. There are many instances of unclear sentence structure. We suggest that you engage a copy editor to review your manuscript before resubmitting it.

Reviewers' comments:

Reviewer's Responses to Questions

**Comments to the Author**

1. If the authors have adequately addressed your comments raised in a previous round of review and you feel that this manuscript is now acceptable for publication, you may indicate that here to bypass the “Comments to the Author” section, enter your conflict of interest statement in the “Confidential to Editor” section, and submit your "Accept" recommendation.

Reviewer #1: All comments have been addressed

2. Does this manuscript meet PLOS Global Public Health’s publication criteria? Is the manuscript technically sound, and do the data support the conclusions? The manuscript must describe methodologically and ethically rigorous research with conclusions that are appropriately drawn based on the data presented.

Reviewer #1: Yes

3. Has the statistical analysis been performed appropriately and rigorously?

Reviewer #1: N/A

4. Have the authors made all data underlying the findings in their manuscript fully available (please refer to the Data Availability Statement at the start of the manuscript PDF file)?

Reviewer #1: Yes

5. Is the manuscript presented in an intelligible fashion and written in standard English?

Reviewer #1: No

6. Review Comments to the Author

Reviewer #1: All substantial comments have been addressed. Congratulations on a solid paper.

However, the manuscript still contains several typos, including misplaced commas; below is a sample of several which I saw, but it is by no means an exhaustive list. I am including this because PLOS GPH does not copy-edit accepted manuscripts.

Methods: “An specific”  "a specific"

Results:

Direct impact: “There were no clear indications were” – “No clear indications were provided” is sufficient.

As  like

WhatsaApp –> WhatsApp

Midwive – use either singular “midwife” or plural "midwives"

Conclusions:

“Control de pandemic” ?? Should this be "pandemic control?"

7. PLOS authors have the option to publish the peer review history of their article (what does this mean?). If published, this will include your full peer review and any attached files.

**Do you want your identity to be public for this peer review?** For information about this choice, including consent withdrawal, please see our Privacy Policy.

Reviewer #1: **Yes: **Jason Johnson-Peretz

---

## [Editor Report · Decision Letter 2]

5 Mar 2024

PGPH-D-23-01179R2

ENSURING SEXUAL AND REPRODUCTIVE HEALTHCARE SERVICES AMIDST A PANDEMIC: EXPERIENCES FROM HEALTH WORKERS IN LIMA, PERU

Dear Dr. Gianella Malca,

Thank you for submitting your manuscript to PLOS Global Public Health. After careful consideration, we feel that it has merit but does not fully meet PLOS Global Public Health’s publication criteria as it currently stands. Therefore, we invite you to submit a revised version of the manuscript that addresses the points raised during the review process.

Thank you very much for addressing the copy editing concerns on your manuscript. Before we proceed to a final decision on your manuscript, a few concerns were noted that should be addressed:

1) The quality of reporting in the manuscript could be improved in places. For example, details about the inclusion criteria and recruitment methods are a little thin. Please complete and upload a copy of the COREQ or other appropriate reporting checklist for qualitative studies with your revision.

2) Please ensure that your conclusions do not over-generalize, since your study only looked at sexual/reproductive health workers and results cannot be directly applied to health workers in all areas of health care.

3) In your Introduction you state "little is known about the impact these COVID-19 related factors on the health workforce and their activities in continuing to deliver primary health", however, several studies have discussed the impact of COVID-19 on the health workforce. Therefore, please ensure you have sufficiently discussed the related literature within this topic and highlight how in relation to those studies, the current manuscript further contributes to scientific knowledge.

We look forward to receiving your revised manuscript.

Kind regards,

Marianne Clemence

Staff Editor

Journal Requirements:

2. We have noticed that you have uploaded Supporting Information files, but you have not included a list of legends. Please add a full list of legends for your Supporting Information files after the references list.
---

## [Editor Report · Decision Letter 3]

12 Apr 2024

ENSURING SEXUAL AND REPRODUCTIVE HEALTHCARE SERVICES AMIDST A PANDEMIC: EXPERIENCES FROM HEALTH WORKERS IN LIMA, PERU

PGPH-D-23-01179R3

Dear Dr. Gianella Malca,

We are pleased to inform you that your manuscript 'ENSURING SEXUAL AND REPRODUCTIVE HEALTHCARE SERVICES AMIDST A PANDEMIC: EXPERIENCES FROM HEALTH WORKERS IN LIMA, PERU' has been provisionally accepted for publication in PLOS Global Public Health.

Best regards,

Julia Robinson

Executive Editor